# Land Use Change and Its Impact on Landscape Ecological Risk in Typical Areas of the Yellow River Basin in China

**DOI:** 10.3390/ijerph182111301

**Published:** 2021-10-28

**Authors:** Yanbo Qu, Haining Zong, Desheng Su, Zongli Ping, Mei Guan

**Affiliations:** 1School of Public Administration and Policy, Shandong University of Finance and Economics, Jinan 250014, China; zhn202109034@mail.sdufe.edu.cn (H.Z.); 20170927328@mail.sdufe.edu.cn (D.S.); 2Shandong Institute of Territorial and Spatial Planning, Government-Affiliated Institution, Jinan 250014, China; zwy212109016@mail.sdufe.edu.cn (Z.P.); ll212109037@mail.sdufe.edu.cn (M.G.)

**Keywords:** land use, landscape ecological risk, ecological risk assessment, influencing factors, Yellow River Basin

## Abstract

The basic premise of regional ecological construction would be to scientifically and effectively grasp the characteristics of land use change and its impact on landscape ecological risk. The research objects of this paper are the typical areas of the Yellow River Basin in China and “process-change-drive” as the logical main line. Moreover, this paper is based on multi-period land use remote sensing data from 2000 to 2020, the regional land use change process and influencing factors are identified, the temporal and spatial evolution and response process of landscape ecological risk are discussed, and the land use zoning control strategy to reduce ecological risk is put forward. The results indicated: (1) The scale and structure of land use show the characteristics of “many-to-one” and “one-to-many”; (2) the process of land use change is affected by the alternation of multiple factors. The natural environment and socio-economic factors dominate in the early stage and the location and policy factors have a significant impact in the later stage; (3) the overall landscape ecological risk level and conversion rate show a trend of “high in the southeast, low in the northwest”, shift from low to high and landscape ecological risks gradually increase; and (4) in order to improve the regional ecological safety and according to the characteristics of landscape ecological risk and spatial heterogeneity, we should adopt the management and control zoning method and set different levels of control intensity (from key intensity to strict intensity to general intensity), and develop differentiated land use control strategies.

## 1. Introduction

Ecological environmental risks have gradually become an important factor affecting national security and restricting the sustainable development of the economy and healthy society [1,2]. The continuous implementation of the “Five Development Concepts” and the “Two Mountains Theory” [3], as well as analyzing and resolving ecological risks in time in order to guarantee the environmental safety as a crucial part of achieving ecological civilization in China based on its construction goals, include harmonious symbiosis, virtuous circle, and comprehensive development. Ecological risk refers to the potential damage to the structure or function of the ecosystem, caused by accidents or hazards in the region [4,5]. The relationship between land use change and landscape ecological risk (LER) is complex. The process of land use type change affects a series of ecological processes, such as the atmosphere, soil, water bodies, and organisms. As a result, the ecosystem structure and function are changed by land cover changes, caused by land use changes and the extensive effects of ecological changes [6,7,8]. The landscape ecological risk assessment (LERA), based on the pattern of land use change, can measure the adverse effects of the combined landscape pattern and ecological process, and it is of great significance to analyze the global aspects, dynamic evolution, and optimization of prevention and control risk of regional ecological risks [9,10,11]. The rapid economic development has caused many ecological safety issues to be solved urgently. The protection of ecologically fragile areas has an irreplaceable role in promoting ecological safety and harmony between human and land. Existing studies have mostly carried out beneficial explorations on the LER in small-scale and intense human activities, while less attention has been given to large and medium scale ecologically sensitive and fragile natural areas. The study of land use change and ecological risk response in fragile areas provides a foundation for regional ecological construction, environmental restoration, and high-quality sustainable development [12].

The concept of landscape was first proposed by the German geographer, Carl Troll. As a complex of natural surface, the landscape is a collection of highly spatially heterogeneous regional ecosystems [13]. At present, LER research has gradually shifted from unilateral examination on risk sources and risk receptors to the overall impact of the ecosystem and the spatial correlation of LER, paying more attention to the macroscopicity and practicability [14,15,16]. Scholars have conducted in-depth research on land use and ecological risks, and LER patterns assessment based on the perspective of land use change, which has gradually become the mainstream of research [17]. At the beginning of the 1990s, scholars such as Heggem et al. [18], Kapustka et al. [19], and Estoque et al. [20] introduced landscape pattern analysis methods to assess the impact of human activities on ecological changes in the watershed. Most of the studies use high-risk communities as the evaluation unit, disregarding the spatial differences and natural geography relation, which is not helping in grasping the overall landscape pattern. However, the geographical significance of the evaluation cannot be ignored. As the evaluation unit, the administrative area should be ideal for studying the prevention and risk control in sensitive and fragile areas, in order for different policies to be defined for the ecological risk situation of different administrative regions [21,22]. The research on LER, brought by land use changes, is generally established on two evaluation models. One is based on the traditional “source-receptor analysis, exposure evaluation, and hazard evaluation” inherent mode. The ecological risk assessment index system is constructed from the risk source intensity, the receptor exposure, and the risk effect. The other model directly evaluates the LER from the landscape pattern and uses the landscape ecological index from the perspective of landscape ecology to reflect the ecological effect of land use and land cover change (LULCC) [23,24,25,26]. Comparing these two models, the conclusion is that the model based on the landscape perspective is more suitable for evaluating the ecological risks caused by human activities, since human activities will have an immense influence on the landscape pattern. The consequences of this influence will directly lead to changes in the ecological environment, and the research in this area becomes more reliable. Many academics have conducted research, analysis, and experiments on various landscape indices [10,27,28]. Therefore, based on the landscape pattern, the ecological risk assessment is relatively scientifically founded and feasible [29]. Existing studies mostly focused on the construction of LER models and spatial analysis [30,31]. The lack of attention to the LER and land use change process and the lack of time-period process research have led to a decline in the credibility and applicability of the risk assessment results. At the same time, according to the administrative division, the meso-scale study which takes into account the core field of economic development and the area with fragile ecological environment has gradually become a research hotspot [32]. In summary, scholars mainly explore landscape risks by constructing evaluation models, and have formed relatively mature evaluation methods and systems. They rarely involve a quantitative analysis of the influencing factors or driving forces of land use changes that lead to the differentiation of LER, and lack zoned explorations of local LER control measures [33].

The ecological protection and high-quality development of the Yellow River Basin are related to China’s social development and ecological protection, and have risen to a major national strategy. As the only province along the Yellow River in the East, along the Yellow River and along the coast, Shandong Province is in a dominant position in the “Yellow River Strategy”. The areas along the Yellow River in Shandong Province have the most developed economy and the largest permanent population among the provinces and cities along the Yellow River, with the highest urbanization rate and obvious geographical advantages of river sea intersection. At the same time, since the entire territory is downstream, the degradation of the natural ecological environment has become increasingly prominent, the pressure of ecological protection is great, and the problem of ecological security is prominent. The areas along the Yellow River in Shandong Province are a typical representative of the Yellow River Basin. Studying the landscape ecological risk response and countermeasures in typical areas of the Yellow River Basin is conducive to the healthy and sustainable development of the Yellow River Basin.

Therefore, this article selects the typical areas of the Yellow River Basin that are both ecologically fragile and economically developed as the study area. Taking the county area as the research scale, based on the land use data of the typical areas of the Yellow River Basin in 2000, 2010, and 2020, the landscape pattern index is calculated with the help of FRAGSTATS 4.2 (a program designed to compute a wide variety of landscape metrics for categorical map patterns) to construct the LERA model. It depicts the process of land use change, based on multiple dimensions of “process-change-drive” [34], explores and analyzes the influencing factors of land use change, fully reveals the temporal and spatial differentiation and LER transfer laws, explores the LER response of land use changes and conducts zoned prevention and control, proposing targeted management and control recommendations, in order to provide a useful reference for the realization of ecological protection and high-quality development strategy in the Yellow River Basin.

## 2. Materials and Methods

### 2.1. Study Area

The typical areas of the Yellow River Basin (34°26′–38°16′ N, 114°45′–119°19′ E) are located in the west of Shandong Province (Figure 1). The Yellow River enters from Dongming County of Shandong Province, flows north to east and through Heze, Jining, Taian, Liaocheng, Jinan, Dezhou, Binzhou, Zibo, and Dongying. The typical areas of the Yellow River Basin are a warm temperate humid and semi-humid monsoon climate type with four distinct seasons, significantly dry and wet, followed by rain and heat in the same season. The total land area in nine cities of seventy-seven counties along the Yellow River is 82,500 square kilometers (km^2^), accounting for 53.4% of the provincial total land area. The terrain is complex, i.e., mountains and hills account for about 35% of the area, while plains, depressions, and beaches account for about 65% of the total area of the Yellow River Basin in Shandong. The mountains in central and southern Shandong are protruding. The northwest of Shandong is low-lying and flat, and the gentle hills in the southwest of Shandong are undulating, forming a general terrain with mountains and hills as the skeleton, while plains and basins are interlaced and ringed in between. In 2020, the regional gross domestic product (GDP) of the nine cities along the Yellow River is CNY 3891.7 billion, accounting for 50.9% of Shandong regional GDP, with a permanent population of 54.272 million. It is an important part of the economic circle for the capital of Shandong Province and southern Shandong. Ecological and environmental problems, such as soil salinization, desertification, and soil erosion in the region, are becoming more serious. Environmental pollution and degradation are prominent, and the LER prevention and control is imminent. Taking this as a case area to carry out LER research has an important significance.

### 2.2. Data

The land use data are from the Resource and Environment Science and Data Center (RESDC) of Chinese Academy of Sciences (http://www.resdc.cn) (accessed on 3 February 2021), including the national land use remote sensing data in 2000, 2010, and 2020 (raster data, resolution of 1 km) and China’s administrative division data (vector data). The land use data were cropped, according to the administrative boundaries of the study field, and the land use data of the typical areas of the Yellow River Basin in 2000, 2010, and 2020 were obtained. The test data with ENVI 5.3 (tool for processing remote sensing images), show the comprehensive accuracy of the three periods over 92%. On this basis, with the help of the ArcGIS operating platform, the relevant data of land use can be extracted from the administrative area of the study area, and construct a land use database of the typical area in the Yellow River Basin. Referring to the existing literature, it can reclassify the relevant data of land use in ArcGIS, and divide the land use types into six categories: Cultivated land, wetland, grass land, forest land, construction land, and bare land [17]. The data in the analysis of influencing factors include relevant data, such as natural environment foundation, social and economic conditions, traffic and location conditions, policy and institutional environment, which are from the geospatial data cloud (http://www.gscloud.cn) (accessed on 5 March 2021), the global nightlight remote sensing data (https://www.nature.com/sdata) (accessed on 5 March 2021), Chinese Soil Database of Nanjing Institute of Soil, Chinese Academy of Sciences (http://vdb3.soil.csdb.cn) (accessed on 5 March 2021), National Geomatics Center of China (NGCC, http://ngcc.sbsm.gov.cn/) (accessed on 5 March 2021), and statistical yearbooks of the nine cities along the Yellow River in 2000, 2021, and 2020. See Table 1 for details:

### 2.3. Methods

Changes in natural factors and human interference, directly or indirectly, affect the land use structure and function, and changes in land use types further lead to changes in landscape patterns [35,36]. Due to the significant landscape pattern heterogeneity and its relation to ecological process, under the stress and involvement of natural background and human geography elements, it will cause potential adverse effects or harms. Moreover, it indicates that LER and land use changes respond to each other [30,37]. LER are caused by land use changes and are driven by natural factors and human activities. To avoid, adapt, and comprehensively manage risks, it is necessary to improve the related factors of land use change and carry out the zoned management and control (Figure 2). Therefore, this paper first applies the chord diagram and geographical information system (GIS) map to identify the overall and process characteristics of regional land use change, and then analyzes the influencing factors and driving mechanism of land use change process with the help of factor detectors in geographic detectors. Second, the LERA model constructed by the landscape disturbance index and landscape vulnerability index is used to explore the temporal and spatial transfer and evolution of LER and the distribution of land types, and the LER response of land use change is obtained with the help of ecological risk contribution rate model. Finally, based on the above research results, it divides the LER management and control area of the typical area of the Yellow River Basin and proposes relevant measures to guide the high-quality sustainable development of the region [38].

#### 2.3.1. Chord Diagram of Land Use Changes

The chord diagram is a graphical method that shows the inter-relationship between data. The data points in the chord diagram are arranged radially in the form of circles, and lines are used to show the connections between the data. The chord diagram can reflect the number of conversions and relationship-flow between different land use types in the process of land use change and visualize it [39]. The wider the chord (connecting line), the higher the number of conversions between land use types. This paper uses the Multi-Charts 1.8 software (https://jshare.com.cn/new) (accessed on 5 March 2021) to visualize the change process of different land use types.

#### 2.3.2. GIS Map Analysis of Land Use Changes

The GIS map analysis is used to reflect the degree of quantitative changes in land use types, and is the basic manifestation of the impact of social and economic activities on land use [40]. The process of land use change reflects the relation of change from one land use type into another, including two conversion directions, transfer-in and transfer-out. The former pays attention to the increase in the transfer-in land use type, and the latter focuses on the reduction of the land use type. This paper uses the grid calculator of ArcGIS 10.3 to superimpose the land use types from 2000 to 2020 (Equation (1)), and obtain the land use change map of the typical area in the Yellow River Basin at different periods as follows:(1)W=A×10+B
where *W* represents the newly generated graph coding; *A* represents the land use atlas unit coding at the beginning of the study; and *B* represents the land use atlas unit coding at the end of the study (secondary classification code). For example, *W* = 12 indicates a GIS map of land use types converted from forest land to wetland.

#### 2.3.3. Land Use Change Influencing Factors Analysis

The geographical detectors can be used to analyze spatial differentiation characteristics and explore the interaction between factors. It is convenient to operate and is less affected by the sample size [41]. The main types of geographical detectors are risk, factor, ecological, and interaction detectors, among which factor detectors can disclose the explanatory power of independent variables to dependent variables [42,43]. This study selects factor detectors, takes typical counties (cities, districts) in the Yellow River Basin as the basic unit, and the six types of land use change rates from 2000 to 2020 as the geographical detector indicators, and carries out an analysis on the influencing factors of LER in typical areas of the Yellow River Basin by GeoDetector (http://www.geodetector.cn) (accessed on 8 March 2021). The method is shown in the following equation:(2)$$q=1-\frac{1}{N\delta^{2}}\overset{H}{\underset{h=1}{\sum }}N_{h}\delta_{h}^{2}$$
where q represents the influencing factors index of land use change; *N* represents the number of global samples; $N_{h}$ represents the number of samples in the secondary region; *H* represents the factors stratification; $\delta^{2}$ represents the total variance of the whole region; and $\;\delta_{h}^{2}$ represents the secondary region discrete variance. The value interval of *q* is [0, 1], and the greater the q value, the greater the influence force on land use change.

Land use change occurs within the three-fold framework of natural system, socio-economic system, and institutional system. The typical areas of the Yellow River Basin are greatly undulating, with rich landform types, strongly affected by the monsoon climate, and the significant change rate of average annual precipitation and average annual temperature. At the same time, with the advancement of the Yellow River Basin regional development strategy, the typical areas in the Yellow River Basin are committed to industrial structure adjustment and upgrading, infrastructure construction, and ecosystem restoration. Government departments provide continuous and strong financial support for the industrial development of the region. In addition, the level of urbanization, industrial structure, quality of life, and ecological environment in the region have been significantly improved. Therefore, based on the actual conditions of the typical areas in the Yellow River Basin, such as significant topographic fluctuations and rapid regional economic development, thirteen indicators were selected as the detection factors of land use changes, including natural environment foundation, social and economic conditions, traffic and location conditions, and policy and institutional environment (Table 2). The ArcGIS is used to rasterize all of the influence factors and unify the projection coordinate system. In addition, the natural break point method is used to process the spatial discretization of the influencing factors, to measure the degree of influence between land use changes, and the influencing factors in typical areas of the Yellow River Basin.

#### 2.3.4. LERA Method

The landscape pattern production is the result of differences of human impact on natural ecosystems. This ecological impact presents regional and cumulative characteristics. The intensity of external disturbance and the ability of internal resistance to the disturbance of the ecosystems, represented by different landscapes, determine the size of LER [44].

In view of the relation between landscape pattern and ecological risk, the landscape disturbance index and landscape vulnerability index are used to construct a LERA model [21,22,23,24,25]. The landscape disturbance index is composed of landscape fragmentation index, landscape separation index, and landscape subdimension index.1.Landscape disturbance index (*Li*)

Li indicates the ability of different landscape ecosystems to resist interference from the outside world and self-recovery. The sensitivity of landscape ecosystems increases with the rise of the landscape disturbance pattern, which leads to a greater LER. By selecting the landscape fragmentation index (*Bi*), landscape separation index (*Si*), and landscape subdimension index (*Fi*), the Li is constructed. Bi represents the degree of fragmentation of the landscape space, from single continuous to complex discontinuous, reflecting the degree of natural or human disturbance to the landscape. It shows that the larger the value, the lower the stability of the corresponding landscape type. Si refers to the degree of separation in different patches of the landscape. The larger the value, the more scattered the corresponding landscape and the more complex the landscape distribution. Fi is an index used to determine the patch shape influence on the internal patch ecological process. The larger the value, the more complex the corresponding patch shape. The indices calculation equations are as follows:(3)$$L_{i}=aB_{i}+bS_{i}+cF_{i}$$
(4)$$B_{i}=\frac{n_{i}}{A_{i}}$$
(5)$$S_{i}=\frac{A}{2A_{i}}\sqrt{\frac{n_{i}}{A}}$$
(6)$$F_{i}=\frac{2ln\left(P_{i}/4\right)}{lnA_{i}}$$
where *a* represents the weight value of *Bi*; *b* represents the weight value of *Si*; *c* represents the weight value of *Fi*; and *a* + *b* + *c* = 1. According to the relevant research results and the actual situation of the study area, the values are assigned to 0.5, 0.3, 0.2 [3,21]; $n_{i}$ represents the number of patches in the *i*-th landscape; $A_{i}$ represents the area of the *i*-th landscape, in km^2^; *A* represents the total area of all the landscapes, in km^2^; and $P_{i}$ represents the perimeter of the *i*-th landscape, in km.2.Landscape vulnerability index (*Wi*)

*Wi* represents the sensitivity of different landscape ecosystems to external disturbances. The larger the value, the lower the stability of the ecosystem and the higher the possibility of damage. Based on previous studies and the actual situation in typical areas of the Yellow River Basin, the six types of landscapes from high to low are assigned as follows: 6—construction land, 5—bare land, 4—cultivated land, 3—grass land, 2—wetland, and 1—forest land [24,25], which are normalized.
3.LERA index ($ERI_{k}$)

$\;ERI_{k}$ is constructed by Li and Wi. It separates the spatial ecological risk from the landscape spatial structure and represents the degree of ecological loss within the assessment unit. The greater the value, the higher the corresponding LER, as shown in the following equation:(7)$$ERI_{k}=\overset{z}{\underset{i=1}{\sum }}\frac{A_{ki}}{A_{k}}\left(L_{i}W_{i}\right)$$
where $ERI_{k}$ represents the LERA index unit *k*; *z* represents the number of landscape types; $A_{ki}$ represents the area of the *i-*th landscape in the LERA unit *k*, in km^2^; and $A_{k}$ represents the area of landscape ecological risk assessment unit *k*, in km^2^.

#### 2.3.5. Land Use Change Ecological Risk Contribution Rate

The ecological risk contribution rate of land use changes refers to the degree of change in the regional LER, which is caused by a certain land use type change [45]. A positive value indicates that this type of change has aggravated the LER in the region, and a negative value indicates that this type of change improves the LER in the region. Isolating the main land use types that affect the LER changes is conducive to exploring the leading factors of changes in regional LER [46]. The calculation is as follows:(8)$$LEI=\frac{\left(ERI_{1}-ERI_{0}\right)LA}{TA}$$
where *LEI* represents the ecological risk contribution rate of land use changes; $ERI_{0}$ represents a LER index at the early stage of change of land use type, and $ERI_{1}$ represents a LER index at the end of change of land use type; *LA* represents the area of the change type; and *TA* represents the total area of the region.

The positive and negative analysis of the LEI can be used to comprehensively determine the land use types that affect the LER change index in typical areas of the Yellow River Basin, which is helpful in distinguishing the leading factors of the improvement and degradation of LER in typical areas of the Yellow River Basin.

## 3. Results

### 3.1. Analysis of the Overall Characteristics of the Land Use Change

From 2000 to 2020, the land use is dominated by cultivated and construction land (Table 3), with the largest proportion of cultivated land in typical areas of the Yellow River Basin. This is consistent with the characteristics of Shandong Province as a major agricultural province and the rapid development of economic society and urbanization. From the perspective of time series characteristics, the area of forest land first increased and then decreased. The fluctuation range in the first 10 years was small, and the area of forest land decreased by 4.41 hectare (ha) in the next 10 years. The area of grass land, cultivated land, and bare land has shown a downward trend in the past two decades. The area of cultivated land has relatively stable changes and it decreased by 3.41%. This is inseparable from the Shandong Province emphasis on cultivated land safety and the effective implementation of basic farmland protection policy. The area of grass land has decreased by 42.11%, indicating that the implementation of the 2003 policy of returning cultivated land to forests and grass land needs to be strengthened. In addition, the area of bare land is significantly smaller, and a lot of bare land is fully utilized for project construction. The area of wetland and construction land is increasing gradually, and the increase rate of wetland, from 2010 to 2020, is as high as 51.86%. This is a remarkable result of paying attention to the protection of wetland system and vigorously investing in the construction of nature reserves and wetland parks in typical areas of the Yellow River Basin. The area of construction land has increased for 33.83%, reflecting the common needs of rapid urbanization and economic development.

### 3.2. Land Use Change Process Analysis

1.Scale feature analysis

From a staged perspective, the scale of land use change from 2000 to 2010 showed the characteristics of “many to one” and “one to many” in typical areas of the Yellow River Basin (Figure 3). Among them, the main construction land transfer sources are cultivated land, bare land, and wetland. The transferred areas are 1144, 314, and 125 km^2^. The main wetland transfer sources are cultivated land, grass land, and bare land, and the transferred areas are 239, 128, and 62 km^2^. The forest land transferred area is relatively small, and the main sources are cultivated land and grass land, and the transfer area is 21 and 17 km^2^, respectively. The construction land, wetland, and forest land transferred area decreases successively, showing a “many-to-one” characteristic. Cultivated land is mainly converted into construction land and wetland, with the transferred area as 1144 and 239 km^2^. Grass land is mainly converted into cultivated land, wetland, and construction land, with the transferred area as 460, 128, and 53 km^2^, respectively. Bare land is mainly converted into construction land, cultivated land, and wetland, and the transferred area is 314, 174, and 62 km^2^, respectively. The transferred area of cultivated land, grass land, and bare land successively decreased, showing the characteristics of “one-to-many”.

Except for bare land, from 2010 to 2020, the dominant characteristics of “one-to-many” transfers continue, mainly for cultivated land, wetland, and construction land, with transfer areas as 828, 412, and 191 km^2^, respectively. Other land use types show balanced conversion attributes, but their respective transfer-in and transfer-out dominant types show significant differences. Among them, the reciprocal conversion of cultivated land and construction land is crucial. The main source of cultivated land transfer is construction land, with an area of 7477 km^2^, while the main source of construction land transfer is cultivated land with an area of 9922 km^2^. Wetland, construction land, and cultivated land are mutually transformed, but the transfer-in scale is larger than the transfer-out scale. Grass land, wood land, and cultivated land are mutually transformed, and the transfer-out scale is larger than the transfer-in scale. Forest land, cultivated land, and construction land are mutually transformed, and the transfer-out scale is not much different from the transfer-in scale. From the perspective of the whole time period, the land use change trend scale in typical areas of the Yellow River Basin, from 2000 to 2020, follows the elements of transformation from 2010 to 2020, and the scale of conversion of some internal land use types slightly increases or decreases.

Here, FL, WL, GL, CL, BL, and CONL represent forest land, wetland, grass land, cultivated land, bare land, and construction land, respectively.2.The spatial map analysis

The land use change map of the typical areas of the Yellow River Basin generates 30 types of map units from 2000 to 2010, and 24 types of map units are changed (Figure 4). Among them, the map type of “cultivated land → construction land” (code 46) is the most obvious, and widely distributed in urban agglomerations, mostly in urban fringe areas, which is in line with the features of rapid urbanization expansion in this region. At times when the government paid insufficient attention to the cultivated land safety, part of the cultivated land was transformed into urban and rural construction land. The “cultivated land → wetland” (code 42) and “grass land → wetland” (code 32) are mainly distributed in the coastal areas of Dongying City. To alleviate the ecosystem vulnerability, large amounts of cultivated land and grass land are converted into wetland. From 2010 to 2020, there are 36 types of land use change map units in typical areas of the Yellow River Basin, while 30 types of map units have changed. The spatial aggregation degree is particularly stronger than in the previous stage, and the coastal area is higher than in the inland area. Among them, the most important is the rapid expansion and wide distribution of “construction land → cultivated land” (code 64). The main reason is that the state gives a great authority to the protection of cultivated land and the implementation of the occupation and compensation balance policy. Substantial construction land is converted into cultivated land, which is conducive to the response to the slogan that the total area of Chinese farmland must remain above the red line of 120 million hectares. The “construction land → wetland” (code 62) and “bare land → wetland” (code 52) are becoming more dominant in the transformation, and they are distributed in the northern coastal areas. In the development theory of transformation, from focusing on speed growth to high-quality development and ecological protection, Shandong Province pays more attention to ecological environment protection, in order to gradually improve the ecological conditions of key protection areas, and the scale of wetland has been greatly expanded.

Regarding the whole period, the land use change map of the typical areas of the Yellow River Basin generated a total of 36 types of map units from 2000 to 2020, and a total of 30 types of map units have changed. The overall change is similar to the spatial distribution of the period from 2010 to 2020. Among them, “cultivated land → construction land” (code 46) and “construction land → cultivated land” (code 64) are more evenly distributed in typical areas of the Yellow River Basin, with obvious spatial dispersion. The “construction land → wetland” (code 62) and “bare land → wetland” (code 52) are mainly distributed in the northern coastal areas, while “cultivated land → wetland” (code 42) is mainly distributed near the Yellow River and its tributaries.

Here, codes 1–6 represent forest land, wetland, grass land, cultivated land, bare land, and construction land, respectively. The map unit represents land use type transformations, and the coding is a combination of secondary classification coding from two transformations, e.g., “forest land **→** wetland” (code 12).

From the fluctuation process point of view, from 2000 to 2020, the types of land use growth in typical areas of the Yellow River Basin are relatively obvious and complex. Among them, the land use was relatively stable from 2000 to 2010, with more than 95% of the area unchanged. The types of land use that have changed are mainly new construction land and cultivated land, which are mainly distributed in offshore areas. Urbanization expansion was the mainstream of the development of the typical areas in the Yellow River Basin during this period. Relying on marine resources, many ports and salt fields were built, while plenty of cultivated lands were supplemented through the saline-alkali land management. By 2010–2020, the overall land use changes are quite drastic, with the change area accounting for more than 35%, mainly cultivated land, construction land, and wetland. The newly added cultivated land is distributed more along the Yellow River banks and at the sea mouth. The newly-added construction land is mainly distributed around the existing urban and town areas, and the newly-added wetland is concentrated in the coastal areas. In this period, a national agricultural high-tech industry demonstration zone was established, and it provides support for the development of ecological and circular agriculture. Soil improvement has encouraged the planting and promotion of salt-tolerant crops such as cotton, vegetables, and forests, which has led to a significant increase in cultivated land. Driven by industrial transformation and development, the urbanization of the population has been brisk, and construction land has also increased. At the same time, ecological protection began to be carried out vigorously. The original overexploited and constructed industrial and mining land and the heavily polluted chemical enterprises gradually withdrew and changed from construction land to wetland.

### 3.3. Analysis of Land Use Change Influencing Factors

Altogether, the spatial differentiation characteristics of land use changes in typical areas of the Yellow River Basin are affected by natural and socio-economic factors, traffic, location, and policy, but the influencing factors and intensity of land use change rates of various types of land show some difference (Figure 5).

From the forest land perspective, the top three influencing factors, leading to the differentiation of forest land spatial change rates, from 2000 to 2010, are elevation (0.67), soil organic matter content (0.53), and change rate of public financial expenditure (0.37). Elevation affects the rate of conversion of other land types to forest land, and soil conditions affect the scale of conversion of forest land to grass land and cultivated land. Since the 2003 policy of farm land to forest return, the tendency of fiscal expenditure has prompted part of the cultivated land to be converted into forest land. During this time, the natural environment and the policy environment were the main influencing factors. Compared with the previous period, from 2010 to 2020, the top three influencing factors have been transformed into elevation (0.57), change rate of urbanization (0.43), and change rate of ground-average agricultural machinery (0.39). With the accelerated urbanization, due to the lax industrial orientation and policy, the construction land has been invaded and occupied by forest land, and the continuous improvement of agricultural technology has promoted the use of forest land that is not suitable for planting crops for food production, which are the leading factor at this stage.

From the perspective of wetlands from 2000 to 2010, the top three influencing factors that led to the spatial differentiation of wetland change rates were average annual precipitation (0.52), public financial expenditure change rate (0.35), and population change rate (0.33). The decreasing trend of precipitation, from coastal to inland territory, significantly affected the spatial distribution of wetland. Government policies, financial support, and human interference have led to the conversion of cultivated land and bare land into wetland. From 2010 to 2020, the top three influencing factors have been transformed into the public financial expenditure change rate (0.65), population change rate (0.43), and per capita social consumer goods sales change rate (0.37). Due to the establishment of the Yellow River Delta High-Efficiency Eco-Economic Zone, the impact of fiscal expenditures sharply increased. With the growth of the total population and per capita consumption level, due to the ecosystem integrity and ecotourism demand, the wetland area is quickly replenished. The level of social and economic development and policy guidance jointly drive the differentiation of wetland change rates.

From the perspective of grass land from 2000 to 2010, the main three factors leading to the spatial differentiation of grass land change rates are average annual precipitation (0.41), elevation (0.39), and soil organic matter content (0.32). The natural world plays a leading role in the transformation between grass land and forest land and between grass land and cultivated land. During 2010–2020, the three dominant factors are average annual precipitation (0.43), elevation (0.42), and average ground agricultural machinery change rate (0.30). In addition to natural factors, the expansion of agricultural technology promotes the full growth of animal husbandry, and part of grass land is transformed into cultivated land, and these are all key influencing factors.

From the cultivated land perspective from 2000 to 2010, the main three influencing factors leading to the spatial differentiation of cultivated land change rates are urbanization change rate (0.58), average annual precipitation (0.51), and population change rate (0.43). Driven by the interests of urbanization, a large number of cultivated lands are converted to construction land. Regional precipitation conditions are restraining the cultivated land expansion scale. The demand for homesteads, due to the population increase, will appropriately reduce the scale of cultivated land appropriately. Compared with the previous period, the change rates of average ground agricultural machinery (0.42), average ground investment in fixed assets (0.39), and the town center distance (0.36) have become the dominant factors. The improvement of the agricultural technology accelerates the agricultural production efficiency, but also creates favorable conditions for the development of bare land, which is conducive to timely cultivated land renewal. The “Development Plan for the Yellow River Delta High-Efficiency Ecological Economic Zone” initiates a strategic layout for establishing a high-efficiency eco-agricultural demonstration zone, and a further division of variety of production land zoning, differentiated policy support, and capital investment, based on the development characteristics and regional location advantages.

From the bare land point of view from 2000 to 2010, the leading three influencing factors that cause spatial differentiation of the bare land were the change rates of urbanization (0.65), average ground agricultural machinery (0.54), and road density (0.36). The urbanization expansion forces various types of land to be supplemented in time, and technology and transportation advantages provide excellent conditions for the effective development and utilization of bare land. Compared with the previous period, the top three influencing factors are transformed into the average ground agricultural machinery change rate (0.57), the town center distance (0.49), and the average ground fixed assets investment change rate (0.38). As the available land area is declining, location advantages and national investment support policies are conducive to the conversion of bare land to construction land and wetlands that are beneficial to ecological protection, which have become the leading factors.

From the construction land perspective from 2000 to 2010, the dominant three factors leading to the spatial differentiation of the construction land change rate are the urbanization change rate (0.61), the population change rate (0.53), and the road density change rate (0.46). The Yellow River Basin typical area is rich in oil and salt resources, and has an advantage of location transportation. With the level of urbanization and population agglomeration, the demand for industrial, mining, and residential land continues to increase. Generally speaking, the level of ecological environment protection lags behind the urbanization development, and the rapidness of urbanization has aggravated the regional LER. In comparison with the previous period, the main three factors have been transformed into the average ground fixed assets investment change rate (0.65), the town center distance (0.45), and the urbanization change rate (0.43). With the urbanization acceleration, relying on the advantages of industry and location, the metropolitan areas of Jinan, Dongbin, and Jihe formed gradually. At the same time, the Agricultural High-Tech Industry Demonstration Area of the Yellow River Delta was established in 2015, and the Yellow River Delta Industrial Investment Fund provides strong financial support for economic and technological development zones and typical industrial parks.

In view of the influencing factors of land use changes in the two periods, and based on the main three influencing factors, Figure 6 presents the comparative analysis of the driving mechanism in two periods (from 2000 to 2010 and from 2010 to 2020).

During 2000–2010, a land use change pattern, affected by large-scale land development and high-intensity use, mainly resulted from the expansion of construction land, wetland, and forest land. On the one hand, the complex terrain and abundant rainfall provide a good natural basis for the changes in various land types in typical areas of the Yellow River Basin. At the same time, economic development and large-scale construction of transportation infrastructure, such as railways and highways, provide conditions for the mutual circulation of production factors [42]. During this period, the continuous industrialization, urbanization, and population expansion promoted the formation of more urban industrial, mining, and residential land. In addition, the country has introduced comprehensive and diverse policies for returning farm land to forest and grass land and other favorable ecosystems, prompting the expansion of forest land and wetland. From 2010 to 2020, the state began to pay attention to the importance of ecological protection in the Yellow River region, and successively established the Yellow River Delta High-Efficiency Eco-Economic Zone and the Agricultural High-Tech Industrial Demonstration Zone of the Yellow River Delta. The determination of the efficient eco-economic development promotes the corresponding changes of land use in this stage. In one way, affected by the increase in precipitation, the carbon and water cycle within the region has been accelerated. Additionally, with the continuous development of soil improvement technology, a large number of beaches have been developed and utilized, and the scale of new wetland and construction land is relatively large. In another way, the location advantages are prominent at this stage. The types of regional land use are constantly adjusted according to the location, and the industrial structure is more reasonable. Urbanization has shifted from incremental expansion to stock revitalization. At the same time, the investment of a large number of special funds and advances in technology provide economic support for land use changes. This has advanced the modern and efficient agricultural development and the replacement of old growth drivers with new ones, leading to a significant increase in the level of intensive land use in typical areas of the Yellow River Basin.

### 3.4. The LER Spatiotemporal Evolution Analysis

Through the LERA model, the value of each risk unit in 2000, 2010, and 2020 is calculated. Based on the natural break point method, the ecological risk is divided into five levels, corresponding to five levels of risk areas: Low (ERI < 0.36), relatively low (0.36 < ERI < 0.50), medium (0.50 < ERI < 0.58), relatively high (0.58 < ERI < 0.60), and high risk area (ERI > 0.60). Using GIS, the temporal and spatial evolution of risk conversion rate from 2000 to 2010, risk conversion rate from 2010 to 2020, and total conversion rate from 2000 to 2020 are plotted. It can be seen from Figure 7 that the temporal and spatial differentiation of LER in typical areas of the Yellow River Basin are significant. In 2000, high-risk areas accounted for 9.1%, mainly distributed in the central city of Jinan, the Dawen River basin, and Dongying area in northern Shandong. At the same time, taking the high-risk area as the center, the risk level shows a decreasing trend. In 2010, the high-risk area accounted for 7.8%, and the LER level of northern Shandong coastal area and the Dawen River basin decreased. In 2020, the proportion of high-risk areas increased to 14.3% and concentrated in Jinan metropolitan area. The overall LER of Northern Shandong coastal area continued to improve and the effect was significant, gradually showing the trend of “high in southeast, low in northwest” and “high in center, low around”.

Based on the natural break point method, the LER conversion rate is divided into six levels, as shown in Figure 8. From 2000 to 2010, about 59.74% of the area had increased LER. Areas with excellent ecological transformation are distributed in the Dezhou Plain Ecological Zone in the northwest of Shandong, the Jihe Plain Ecological Zone in the southwest of Shandong, the Dongying Coastal Ecological Zone in the north of Shandong, and the Zibin Mountain and hills Ecological Zone in the middle east of Shandong. From 2010 to 2020, about 62.34% of the areas have increased LER, and the area of excellent ecological transformation areas has decreased. The overall landscape ecological transformation rate shows a decreasing trend from southeast to northwest. The landscape risks in the southeast have intensified, and the situation is concerning.

In summary, the conversion rate of LER in typical areas of the Yellow River Basin has gradually increased from 2000 to 2020. The highest value of the positive conversion rate of LER at the county scale has increased from 9.32% to 10.27%, and the negative conversion rate of LER has increased from 2.35% to 19.81%. In terms of spatial distribution, the high positive risk conversion areas are distributed stably and concentrated in the Jinan metropolitan area. The central urban area conversion rate is higher than the surrounding areas. The rapid urbanization and economic development of these areas have led to the continuous expansion of construction land, increasing the ecological pressure, and the LER index has crucially changed. During the study period, the LER in the typical areas of the Yellow River Basin continued to deteriorate. The overall landscape risk in the central urban area increased, the landscape ecology in the northwest region gradually improved, and the ecological risk index showed a decreasing trend, while the ecological risk index in the southeast showed an increasing trend.

By superimposing the LER distribution maps in 2000, 2010, and 2020, the LER transfer levels in typical areas of the Yellow River Basin are obtained (Figure 9). From 2000 to 2010, the risk transfer ratio was 59.74%, among which, the proportion of low risk to relatively low risk, low risk to medium risk, and medium risk to relatively low risk is larger in the number of administrative units. In the first 10 years, the LER levels in typical areas of the Yellow River Basin mostly shifted to relatively low and medium risk. Deterioration was accompanied by improvement. Altogether, the main development trend is a further increase in risk levels. From 2010 to 2020, the risk transfer ratio is 70.13%, among which, the proportion of relatively low risk to relatively high risk, relatively low risk to medium risk, and medium risk to low risk is larger in the number of transferred administrative units. In the next 10 years, the LER of the typical areas of the Yellow River Basin mostly turned to medium-high risk, and the overall risk increased. It can be seen that the LER mostly shifted from low-level to high-level, and the ecological risks are aggravated, which lead to consequential ecological challenges.

Here, A, B, C, D, and E represent low risk area, relatively low risk area, medium risk area, relatively high risk area, and high risk area in 2000, respectively; A1, B1, C1, D1, and E1 represent low risk area, relatively low risk area, medium risk area, relatively high risk area, and high risk area in 2010, respectively; and A2, B2, C2, D2, and E1 represent low risk area, relatively low risk area, medium risk area, relatively high risk area, and high risk area in 2020, respectively.

### 3.5. Analysis of LER Response of Land Use Change

#### 3.5.1. The Relationship between Land Use and LER Conversion

The ArcGIS statistical tools were used to obtain the area proportions of different types of land in the transfer of various levels of ecological risk from 2000 to 2020 (Table 4). In the first decade, forest land was mainly distributed in low positive risk conversion areas, accounting for 27.86% of the total forest area. In the second decade, forest land was mainly distributed in low negative risk conversion areas, accounting for 30.54%. The results show that the LER of the forest land is decreasing and the speed is relatively stable. The ecological improvement of forest land is closely related to the state policy on the cultivated land conversion to forest and forest land protection. The forest land ecological improvement is inseparable from the state policies of returning farm land to forest and forest land protection. From 2000 to 2010, wetlands were mainly distributed in low positive risk conversion areas, accounting for 30.05% of the total wetland area. From 2010 to 2020, wetlands areas in relatively high positive risk conversion regions increased significantly, with a total proportion of 33.96%, indicating that the LER of wetland increased sharply during the study period. In recent years, the excessive development of construction land has had a considerable impact on the wetland system. From 2000 to 2010, grass land was mainly distributed in low negative risk conversion areas, accounting for 34.13% of the total grass land area. From 2010 to 2020, the ratio of grass land in relatively high positive risk conversion areas increased significantly, accounting for 29.43%, indicating that the LER of grass land has increased drastically during the study period, but the proportion of the area distributed in the high risk conversion area is smaller. From 2000 to 2010, cultivated land was mainly distributed in low negative risk conversion areas, accounting for 36.35% of the total cultivated land areas. From 2010 to 2020, the ratio of cultivated land in negative risk conversion areas increased significantly, of which low negative risk accounted for 45.47%. This suggested that the LER of cultivated land has been continuously reduced during the research period, which is closely related to provincial emphasis on ensuring the safety of cultivated land and reducing the human interference.

From 2000 to 2020, the distribution areas of construction land and bare land are mainly positive risk conversion areas, and the LER index crucially increased. Among them, the construction land is mainly distributed in high positive risk conversion areas, which is caused by the demand of rapid economic growth and accelerated urbanization process in typical areas of the Yellow River Basin, and the continuous expansion of the construction land is at the cost of the bare land prosperity and the existing land renewal.

#### 3.5.2. The Impact of Land Use Type Conversion on LER

The regional LER often has two opposite trends of improvement and deterioration at the same time. In a certain area, the two trends can be offset to different degrees. On the one hand, the change of LER index reflects the overall LER development level. On the other hand, the LER stability does not mean that the internal ecological risk has not changed. The process of land use change affects the regional LER differentiation, and the mutual transformation between the different types may have a positive or negative effect on LER.

According to the actual situation of the study area, based on the related research results [47,48], the change in the regional LER index caused by the change of a certain land use type is obtained through the ecological risk contribution rate of land use change, and then the impact of land use type transformation on the LER of the typical areas of the Yellow River Basin was determined. Table 5 shows the change of LER index and its main land use change types contribution rate that promote the improvement and degradation of LER in typical areas of the Yellow River Basin during 2000–2020.

From 2000 to 2010, land use changes generally promoted the increase of the regional LER index. The conversion of cultivated land to construction land was the leading factor in the deterioration of LER at this stage, accounting for 39.82% of the positive effect of LER. Due to the inadequacy of government policies and the demand for rapid economic development, a large amount of cultivated land has been converted to construction land, which has caused a rise in the LER index and substantial deterioration in ecological safety. At the same time, the conversion of wetland to construction land and the conversion of wetland to cultivated land also contributed to the deterioration of LER to a certain extent, and both accounted for nearly one-third of the LER contribution rate of positive effects. Second, the types of land use changes that worsen the LER are relatively concentrated. One is that other types of land use are converted to construction land and the other is that other types of land use, with ecological protection functions, are converted to cultivated land. This is mainly the construction and cultivated land transfer that accounted for more than 92% of the total contribution rate of the positive effects of LER. On the contrary, the transfer of wetland, the utilization of bare land, and the return of farm land to forests and grass land are important factors to improve the LER in the study area. Among them, the occupation of cultivated land and bare land by wetland is dominant, which accounts for more than 58% of the contribution rate of the LER positive effect.

From 2010 to 2020, the conversion of bare land to construction land is the leading factor in the deterioration of LER in typical areas of the Yellow River Basin at this stage, accounting for 37.98% of the LER positive effect. Under the background of urban expansion, the urgency of social and economic development makes a large number of bare lands exploited and utilized. However, due to the lack of institutional guidance and ecological considerations, the large-scale development of bare land leads to the increase of LER index. The rapid urbanization in this period was at the expense of the environment. At the same time, due to the increase in demand for construction land, some cultivated land and wetland have also been converted to construction land which, to a certain extent, has also contributed to the deterioration of the LER in the study area. The two accounted for more than one-third of the LER positive effect, and the ecological safety problem should not be underestimated. During this period, the type of land use change that worsened the LER was the transfer of construction land and cultivated land, and the contribution rate of forest land and grass land occupied by cultivated land slightly increased. The important factors for the LER improvement in the study area are the conversion of construction land to cultivated land and bare land, and the transfer of construction land to wetland, with a contribution rate of more than 70%. The transfers of cultivated land, wetland, and grass land are important reasons for the LER improvement.

Changes in land use types will change the landscape structure and vulnerability index, leading to original landscape fragmentation and increasing landscape ecological risks. In summary, LER improvement and deterioration coexist in typical areas of the Yellow River Basin, but the overall deterioration trend is greater than the improvement trend, and the degree of deterioration of LER continues to increase.

## 4. Discussion

### 4.1. The Relationship between Land Use Change and LER

Land is the carrier of the main social and economic activities, and an important part of the global environmental change and sustainable development research. Driven by economic and social changes and innovation, the types of regional land use have also changed. Land resources are the basis for the survival and development of human society [3]. In recent years, with the acceleration of urbanization, the loss and fragmentation of cultivated land have become more critical. Land itself is the macroscopic representation of the surface landscape, while frequent human activities and high-intensity development and construction make the landscape fragmented and complex, threatening the harmony of humans and land relationship [9]. Changes in land cover, caused by changes in land use, have created changes in the structure and function of ecosystems, and the pattern of surface landscapes has continued to change. Ecosystems may have adverse effects under the direct or indirect effects of land use, and their impacts involve a series of ecological processes such as the atmosphere, soil, water bodies, and organisms and have a wide range of ecological effects. This leads to a variety of real or potential LER, including land degradation [21]. Land use change is highly correlated with the temporal and spatial distribution and dynamics of LER. In fact, the land use change induces LER. To sum up, the LER based on land use change refers to the possibility of changes in landscape structure and reduction of corresponding ecological functions caused by land use and its changes.

LER management refers to the effective prevention and governing measures taken by risk managers for early warning, response, and restoration of LER according to the differences in risk levels and land use types changes in the process of LERA, in order to avoid and reduce LER [44]. The results of LERA of land use can enable managers to understand the spatial distribution of regional LER, identify high risk and medium-high risk areas, and put forward feasible risk control strategies for the temporal and spatial differentiation of different land use types. Land use management is an important approach to LER management. By optimizing the types of land use changes and spatial layout, the regional LER can be effectively reduced. At the same time, land use LERA and management can be continuously improved through mutual feedback. Land use LER management has stages and timeliness. In addition, it is connected with the dynamics of land use research and can promote a virtuous circle of the evaluation process. Land use LER management is a very important part of the ecological risk assessment process. The LERA results can be combined with complex factors, such as regional laws, politics, society, and economy, and its management results can be used for the next landscape risk. Since land use change is driven by nature, social economy, transportation, location, and policies, the LER management of land use can be carried out based on the results of LERA and the factors affecting land use change [48]. Therefore, through the LER differences based on the type of land use, according to the response of the regional LER to the type of land use change, the prevention and control strategies for the LER response and restoration can be proposed.

An important approach for ecological restoration is the construction of ecological projects, most of which involve changes in land use. Carrying out land use LERA provides a strong support and guarantee for ecological restoration. As a new field and important branch of LER research, the LERA of land use can provide a scientific basis and strong decision support for spatial planning and ecological restoration under the background of ecological civilization construction.

### 4.2. Land Use Control Strategy to Reduce Landscape Ecological Risk

The spatial difference in LER is large in typical areas of the Yellow River Basin. Comprehensively considering that the LER grade and conversion rate in the study area show the trend of “high in the center, low around”, the idea of core-peripheral management and control zoning is proposed. Based on the results of county-scale ecological risk diagnosis, in accordance with the need for ecological risk prevention and control and for the convenience of regional management, the typical areas of the Yellow River Basin are divided into “two districts and six pieces” LER key control area, strict control area, and general control area according to the principle of not crossing the municipal administrative region and changing the LER index conversion rate. The “two districts and six pieces” include the core area along the Yellow River and the peripheral linkage area. The core area is divided into four control areas, and the peripheral linkage area is divided into two areas (Figure 10).1.The control strategy of the core area

The core area covers the main stream of the Yellow River and counties (cities and districts) where Dawen River flows, covering an area of 42,800 km^2^. Its development guideline is to build a core area and demonstration leading area for the ecological protection and high-quality progress of the Yellow River Basin. The focus should be on the latter two and on coordinating the inconsistencies among various types of land use, as well as the land use pattern optimization. The essence has to be on improving the central cities’ influence and radiation, as well as strengthening the role of Jinan metropolitan area in ecological protection and high-quality development.

The LER of the strict control area in the upper section belongs to the relatively low risk level and medium risk level, and the conversion rate is first reduced and then increased, while the overall conversion rate is positive. Its development orientation is the Yellow River Ecological Corridor Construction Demonstration Zone. The main types of land use changes are “wetland-construction land” and “construction land-cultivated land”, which lead to increasing LER in recent years. In the future, the region should pay attention to the implementation of the policy regarding the balance between farmland occupation and compensation. In order to effectively protect wetland, forest land, and other lands from occupation, the ecological security barrier system should be optimized and the radiation ability of Heze metropolitan area to the surrounding areas should be increased.

The LER of the general control area of the Dawen River Basin is dominated by medium and high risks. The risk conversion rate is first increased and then reduced, and the overall conversion rate is negative. Its development is positioned as a long-term unruffled demonstration area of the Yellow River. The main types of land use changes are grass land-wetland and grass land-forest land. Moreover, the intensive use of land reduces the ecological protection function. This region should make full use of mountain landscape and water resources. Relying on the existing nature reserves and tourist attractions, a certain protection zone should be set forth, and the region of adjacent forest land and wetland should be expanded to make them concentrated and contiguous to ensure the ecological environment. Then, the fragmentation of the landscape should be reduced and the anti-risk ability of the ecosystem should be improved.

The LER of the key control area in the middle section is mainly high risk, and the conversion rate showed the simultaneous increase and high positive conversion rate. The main types of land use changes are “cultivated land-construction land” and “bare land-construction land”. The economic expansion leads to ’’the blind expansion’’ of construction land, resulting in the loss of cultivated land, the transition of bare land development, and the destruction of landscape ecosystem structure. This area is a key area leading the high-quality development of the Yellow River Basin. The maintenance of forest and grass land should be strengthened to reduce land loss and improve the stability of the ecosystem. In addition, it is necessary to control the regional population and reduce the occupation of cultivated land resources due to the expansion of construction land, and then rationally develop the unused land and adjust the land use structure based on location factors.

The LER of the general control area in the lower section belongs to relatively low and medium risk, and the conversion rate shows the simultaneous decrease, and the overall conversion rate is negative. The main land use change types are “cultivated land-wetland” and “construction land-cultivated land”. The ecological risk has reduced significantly. The development is oriented towards collaborative protection and developing demonstration zones. We should establish wetland parks and other nature reserves, strengthen the maintenance of forest and grass land around rivers, reduce land loss and water fragmentation, and continue to provide policy and financial support, by exploring regional ecological protection and linkages to development mechanisms, while promoting the integrated development of the cities, such as Dongying, Binzhou and Lijin.2.The control strategy of the peripheral linkage area

The peripheral linkage area includes 42 counties (cities, districts), except the core area. The LER level of the collaborative linkage of the key control area mainly belongs to the low risk, while the conversion rate shows a simultaneous increase and a low positive conversion rate. It is located in the ecological area of the northwest plain of Shandong Province. The types of land use change are relatively scattered, and the proportion of cultivated land to construction land is relatively large. With accelerated urbanization, the construction land continues to spread outward, the ecological load is severe, and the ecological risk continues to rise. As Shandong Province is largely agricultural and the protection and utilization of cultivated land is very important, it is recommended to divide the cultivated land based on quality and implement a policy of returning the farm land to forests on cultivated land of poor quality. In addition, the pressure of construction land expansion should be appropriately eased and shifted from incremental development to stock renewal with intensive and efficient utilization. The level of LER in the collaborative linkage strict control area is mainly medium and high risk, and the conversion rate is first increased and then reduced, while the overall conversion rate is positive. The main land use change types in this area are “wetland-construction land” and “forest land-construction land”. We should adopt the leading role of land and space planning to reduce the human interference intensity, make a gradual development of construction land to go together with stock renewal, and protect forest land and wetland from intrusion. While mobilizing the enthusiasm of economic development, the negative impact on landscape ecology will be minimized, in order to maintain the current trend of ecological risk transformation and further improve the landscape ecology. The peripheral linkage area should focus on the joint protection and governance of environment, the integration of living spaces, and the coordinated development of urban and rural areas, while comprehensively reinforcing the collaborative linkage with the core area.

There are extensive distinctions in resource endowments and industrial development levels in typical areas of the Yellow River Basin. The changes in land use types should be promoted according to regional resource endowment conditions, ecological environment capacity, and positioning of main functions. Moreover, its ecological effects should be considered in the formulation of relevant policies and plans. In the process of land use change, it is necessary to actively adjust the structure and layout of land use to strengthen the self-healing function of the ecosystem.

## 5. Conclusions

Based on the land cover data in typical areas of the Yellow River Basin in 2000, 2010, and 2020, this study conducts a research on landscape ecological risk response and countermeasures of land use change. The main conclusions are as follows:(1)The analysis of land use structure demonstrates that the main types of land use in typical areas of the Yellow River Basin are cultivated land and construction land. The change processes of various land use types are significantly different, showing the characteristics of “many-to-one”, “one-to-many”, and “balanced”. Among them, the scale of forest land first increases and then decreases, the area of wetland and construction land increases sharply, and the areas of grass land, cultivated land, and bare land continue to shrink. In the conversion of different land use types, the exchange of cultivated land and construction land, the transfer of construction land to wetland, and the transfer of bare land to wetland are more prominent, as well as denser in coastal areas and more scattered in inland areas;(2)The process of land use change is affected by the factors of nature, society, economy, location, and policy. Within the first decade, the natural environment, society, and economy played a leading role in land use changes. In the second decade, the influence of natural factors declined, while the influence of location and policy factors increased significantly;(3)The results show that the overall LER grades have the characteristics of “high in the southeast, low in the northwest” and “high in the center, low in the surroundings”. The conversion rate of LER increased gradually, and the spatial distribution showed a decreasing trend from southeast to northwest. Most of the ecological risks have shifted from low level to high level. In recent years, the ecological risks of bare land and construction land have increased severely, which should cause concern;(4)The change of land use type will change the landscape structure and vulnerability index, resulting in the original landscape fragmentation and the increase of LER. The landscape ecological improvement and deterioration coexist in typical areas of the Yellow River Basin, but the general landscape ecological deterioration trend is greater than the improvement trend, and the deterioration degree of the landscape ecological environment is increasing; and(5)According to the results of the diagnosis of county-scale LER and the need of ecological risk prevention and control, the typical areas of the Yellow River Basin are divided into “two districts and six pieces” LER with the key control area, strict control area, and general control area. It is committed to transform the Yellow River Basin in Shandong Province into “Shandong model for ecological protection of the Yellow River Basin and a core growth pole for high-quality development”.

This paper attempts to develop differentiated land use regulation strategies for research areas with different landscape ecological risk levels and regional characteristics from the influencing factors and driving mechanism of land use change for the first time. However, the following contents in the future should get further exploration. First, the LER temporal and spatial characteristics have obvious scale effects, and there may be a certain degree of difference in the research results with various exploration scales. This article is based on the remote sensing of land use (raster data, resolution of 1 km) from 2000 to 2020 obtained from the Resource and Environment Science and Data Center (RESDC) of Chinese Academy of Sciences. The results of this study reflect the LER distinctions and their change process at the macro-level in typical areas of the Yellow River Basin, but it is difficult to accurately describe the risk traits of some local areas or land types with a small area. In order to reflect the change of ecological risk more sensitively, it should be considered to study the scale characteristics of ecological risk change by setting grid cells of different sizes. Second, remote sensing data are not only an effective method for land use change research, especially dynamic monitoring, but also play an important role in analyzing the land use change pattern. Remote sensing is able to obtain a large range of data by virtue of its advantages, such as multiple means of obtaining information, fast speed, and not being blocked by terrain, but it may not be able to fully display the LUCC process. Therefore, the combination of traditional ground survey methods, such as land survey, topographic map query, field visit, and questionnaire survey with remote sensing technology will be the focus of the next step. Moreover, it is an important means to deepen the understanding of land use dynamics. Furthermore, this exploration will help in providing more reasonable suggestions for the high-quality development of the basin, and provide more effective regulation and control strategies for decision makers.

## Figures and Tables

**Figure 1 ijerph-18-11301-f001:**
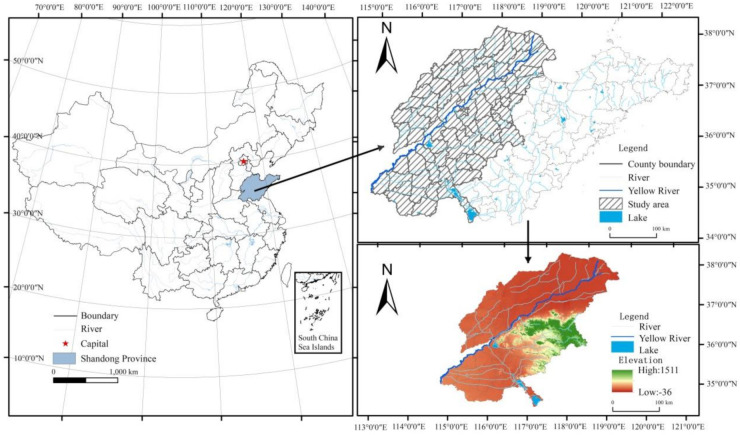
Location of the study area.

**Figure 2 ijerph-18-11301-f002:**
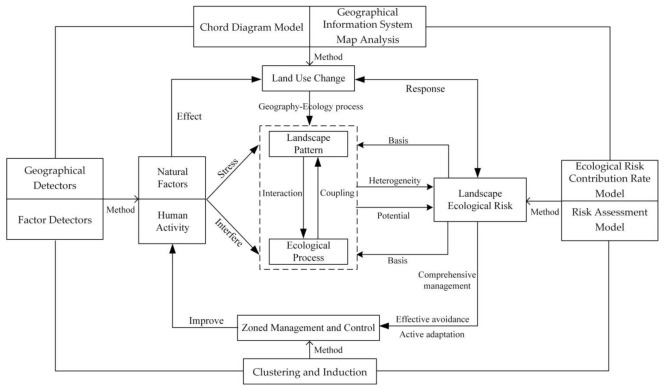
The framework of this study.

**Figure 3 ijerph-18-11301-f003:**
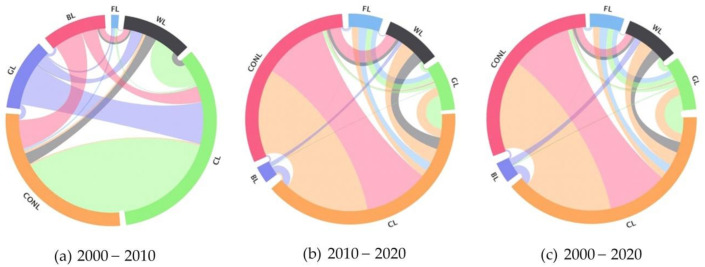
Chord diagram of land use changes in typical areas of the Yellow River Basin from: (**a**) 2000–2010, (**b**) 2010–2020, and (**c**) 2000–2020.

**Figure 4 ijerph-18-11301-f004:**
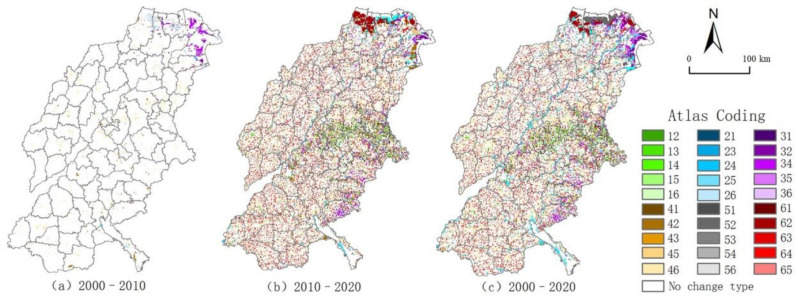
Map of land use changes in typical areas of the Yellow River Basin from: (**a**) 2000–2010, (**b**) 2010–2020, and (**c**) 2000–2020.

**Figure 5 ijerph-18-11301-f005:**
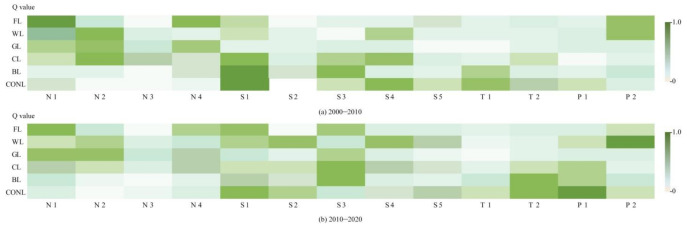
Influencing factors of land use changes: (**a**) From 2000 to 2010 and (**b**) from 2010 to 2020.

**Figure 6 ijerph-18-11301-f006:**
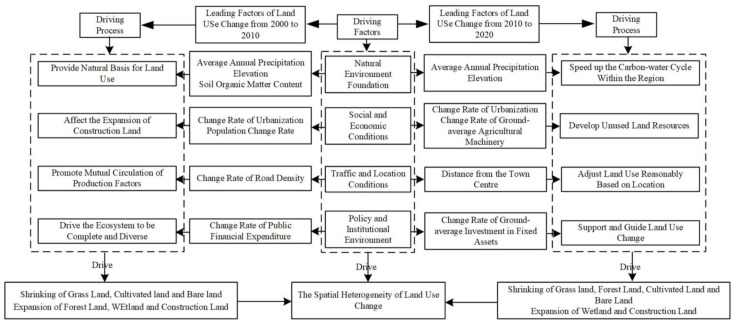
The land use change driving mechanism in typical areas of the Yellow River Basin.

**Figure 7 ijerph-18-11301-f007:**
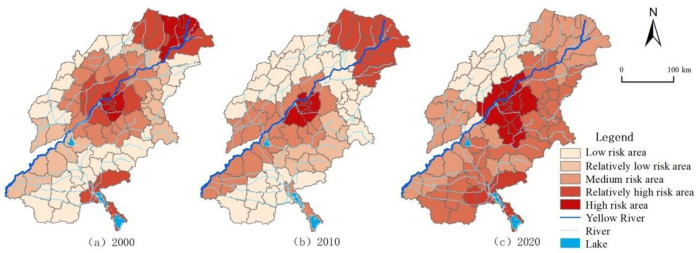
Spatiotemporal variation of LER grade in: (**a**) 2000, (**b**) 2010, and (**c**) 2020.

**Figure 8 ijerph-18-11301-f008:**
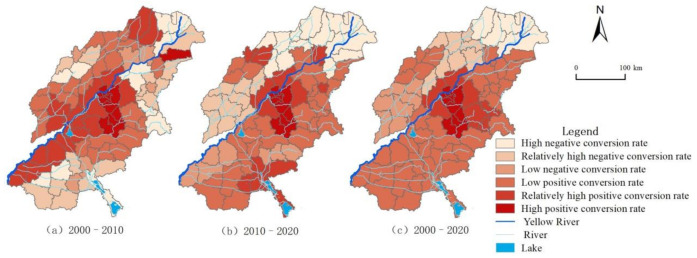
Spatiotemporal variation of LER conversion rate: (**a**) From 2000 to 2010, (**b**) from 2010 to 2020, and (**c**) from 2000 to 2020.

**Figure 9 ijerph-18-11301-f009:**
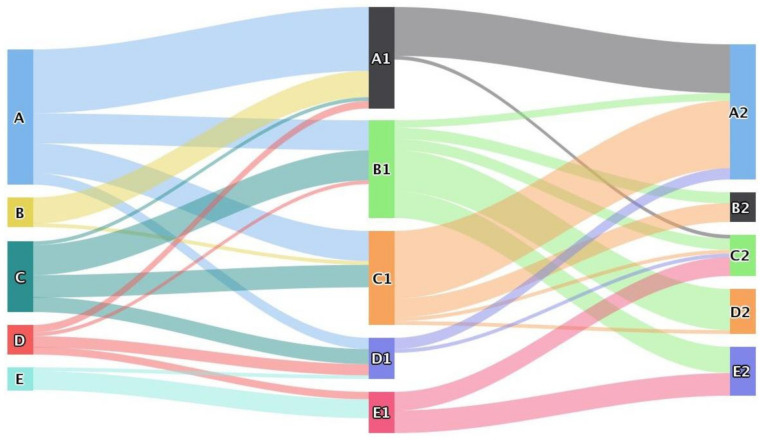
LER transfer situation in typical areas of the Yellow River Basin from 2000 to 2020.

**Figure 10 ijerph-18-11301-f010:**
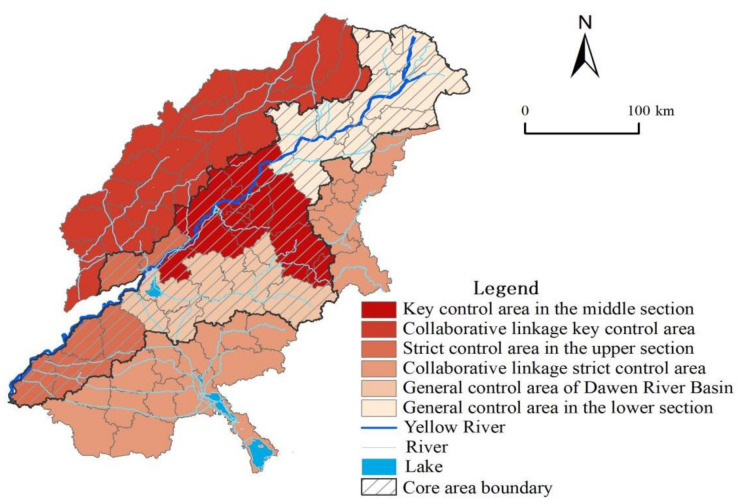
Spatial control distribution of LER in typical areas of the Yellow River Basin.

**Table 1 ijerph-18-11301-t001:** Data sources of land use change influencing factors.

Influencing Factor	Variable	Data Sources
Natural environment foundation	Elevation N1	The geospatial data cloud (http://www.gscloud.cn) (accessed on 5 March 2021)
	Average annual precipitation N2	The Resource and Environmental Science Data Center of the Chinese Academy of Sciences (http://www.resdc.cn) (accessed on 5 March 2021)
	Average annual temperature N3	
	Soil organic matter content N4	Chinese Soil Database of Nanjing Institute of Soil, Chinese Academy of Sciences (http://vdb3.soil.csdb.cn) (accessed on 5 March 2021)
Social and economic conditions	Change rate of urbanization S1	Statistical yearbooks of the nine cities along the Yellow River in 2000, 2021, and 2020.
	Change rate of per capita social consumer goods sales S2	
	Change rate of ground-average agricultural machinery S3	
	Population change rate S4	
	Night light remote sensing S5	The global night light remote sensing data (https://www.nature.com/sdata) (accessed on 5 March 2021)
Traffic and location conditions	Change rate of road density T1	National Basic Geographic Information Center (http://ngcc.sbsm.gov.cn/) (accessed on 5 March 2021)
	Distance from the town center T2	
Policy and institutional environment	Change rate of ground-average investment in fixed assets P1	Statistical yearbooks of the nine cities along the Yellow River in 2000, 2021, and 2020.
	Change rate of public financial expenditure P2	

**Table 2 ijerph-18-11301-t002:** Land use change influencing factors index.

Influencing Factor	Variable	Index Description
Natural environment foundation	Elevation N1	Terrain condition factors
	Average annual precipitation N2	Precipitation condition factors
	Average annual temperature N3	Weather condition factor
	Soil organic matter content N4	Soil condition factors
Social and economic conditions	Change rate of urbanization S1	Development level of urbanization
	Change rate of per capita social consumer goods sales S2	Resident consumption level
	Change rate of ground-average agricultural machinery S3	Level of technological progress
	Population change rate S4	Human-factor level
	Night light remote sensing S5	Level of economic development
Traffic and location conditions	Change rate of road density T1	Traffic accessibility
	Distance from the town center T2	Location advantage degree
Policy and institutional environment	Change rate of ground-average investment in fixed assets P1	Investment level
	Change rate of public financial expenditure P2	Fiscal expenditure level

**Table 3 ijerph-18-11301-t003:** Different types of land use area and change rate in typical areas of the Yellow River Basin from 2000 to 2020.

Land Use Type	Area/km^2^			Change Rate/%		
	2000	2010	2020	2000–2010	2010–2020	2000–2020
Forest land	0.3585	0.3592	0.3151	0.20	−12.28	−12.11
Wetland	0.3786	0.4034	0.6126	6.55	51.86	61.81
Grass land	0.4819	0.4135	0.2790	−14.19	−32.53	−42.11
Cultivated land	5.7116	5.6426	5.5167	−1.21	−2.23	−3.41
Bare land	0.2026	0.1534	0.0401	−24.28	−73.86	−80.21
Construction land	1.1369	1.2986	1.5215	14.22	17.16	33.83

**Table 4 ijerph-18-11301-t004:** Proportion of LER transfer area of different types in typical areas of the Yellow River Basin from 2000 to 2020.

Period	Type	High Negative Risk Conversion Zone	Relatively High Negative Risk Conversion Zone	Low Negative Risk Conversion Zone	Low Positive Risk Conversion Zone	Relatively High Positive Risk Conversion Zone	High Positive Risk Conversion Zone
2000–2010	Forest land	10.35	15.59	20.04	27.86	18.24	7.92
	Wetland	12.85	11.90	16.32	30.05	20.66	8.22
	Grass land	13.56	15.89	34.13	16.34	18.59	1.49
	Cultivated land	10.23	12.45	36.35	15.75	16.34	8.88
	Bare land	7.32	6.34	19.73	20.23	28.96	17.42
	Construction land	8.47	7.40	12.93	21.56	26.45	23.19
2010–2020	Forest land	12.86	18.34	30.54	18.95	16.21	3.10
	Wetland	11.53	10.69	13.25	24.68	33.96	5.89
	Grass land	12.84	14.63	20.34	18.35	29.43	4.41
	Cultivated land	12.31	14.56	45.47	12.96	10.21	4.49
	Bare land	7.02	8.23	12.05	27.45	29.34	15.91
	Construction land	1.79	8.21	12.76	25.67	26.23	25.34

**Table 5 ijerph-18-11301-t005:** The main types of land use changes that affect LER and their contribution rate to ecological risks.

Mode	2000–2010			2010–2020		
	The Main Types of Land Use Changes	Index Change	Contribution Proportion (%)	The Main Types of Land Use Changes	Index Change	Contribution Proportion (%)
Leading to deterioration of LER	Cultivated land -Construction land	0.01628	39.82	Bare land -Construction land	0.01821	37.98
	Wetland -Construction land	0.01069	15.24	Cultivated land -Construction land	0.00953	23.17
	Wetland -Cultivated land	0.00729	12.13	Wetland -Construction land	0.01083	10.86
	Bare land -Construction land	0.00217	11.65	Grass land-Cultivated land	0.00603	6.39
	Forest land -Construction land	0.00372	8.24	Wetland -Cultivated land	0.00527	2.38
	Grass land—Cultivated land	0.00598	5.67	Forest land-Cultivated land	0.00386	1.28
	Total	0.04613	92.75	Total	0.05373	82.06
Leading to the improvement of LER	Cultivated land -Wetland	−0.01023	39.11	Construction land -Cultivated land	−0.00289	36.10
	Bare land -Wetland	−0.00253	18.94	Bare land -Wetland	−0.00363	28.34
	Bare land -Cultivated land	−0.00162	14.83	Construction land -Wetland	−0.00928	10.04
	Grass land-Wetland	−0.00115	9.34	Bare land -Cultivated land	−0.00135	8.96
	Cultivated land -Grass land	−0.00102	5.23	Cultivated land -Wetland	−0.00296	6.48
	Cultivated land -Forest land	−0.00154	3.72	Cultivated land -Grass land	−0.00423	3.27
	Total	−0.01809	91.17	Total	−0.02434	93.19

## Data Availability

All of the relevant data sets in this study are described in the manuscript.

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
