# Peer review of "Land Use Change and Its Impact on Landscape Ecological Risk in Typical Areas of the Yellow River Basin in China"

_ijerph, 2021, doi:10.3390/ijerph182111301_

Round 1
Reviewer 1 Report
The assessed paper presents interesting research on the influence of various factors on the shaping and changes in land use.
It is certainly an important voice in the discussion on the influence of anthropogenic factors on changes.
The authors carried out interesting research procedures and presented proposals for measures to protect the areas at risk.
The selection of the variable parts and their characteristics may be a debatable element. While the level of mechanization of agriculture is an important factor, I am not convinced that it reflects the level of scientific progress. It seems that in this respect it can only be an indicator of technical progress.
The above, however, is only part of the scientific discussion and does not affect the quality of the presented paper.
Reviewer 2 Report
Thank you for the opportunity to read the paper “How to Understand and Deal with Land Use Change and Its Impact on Landscape Ecological Risk in Typical Areas of the Yellow River Basin in China”. They found several predictors while aiming to describe the relationship between Land Use Change and Landscape Ecological Risk in the Yellow River Base, China. The paper is well detailed and written, facilitating the readers understanding of the message. I would recommend simple changes to:
(i) Title. To cut ‘How to Understand and Deal with’ and keep the rest. Justification: when we aim to ‘understand’ we usually select a case study research design (Blaikie and Priest 2019). In this sense, but not exclusively, case studies are useful to undertake deeper insights from a particular social phenomena. This research aims to describe while selecting a deductive approach, thus making no sense to keep ‘understand’ in the title;
(ii) Abstract. To include paragraph from page 5 in the abstract ‘This paper firstly identifies the characteristics of land use change 183 and analyzes its influencing factors in typical areas of the Yellow River Basin, and then explores the temporal and spatial transfer and evolution of LER and the distribution of land types, and the LER response of land use change is obtained. Finally, based on the above research results, it divides the LER management and control area of the typical area of the Yellow River Basin and proposes relevant measures to guide the high-quality sustainable development of the region’, while better summarizing your findings. As it stands it too much information, thus it would be preferable to make it shorter and simpler;
(iii) data and methods. To include and explain your research design (See Blaikie and Priest 2019; Bryman 2017).
Good luck with your research!
Reviewer 3 Report
The paper conducts an interesting research on landscape ecological risk response and countermeasures of land use change. However, there are still some aspects that can be refined. Authors are advised as follows.
It is better to avoid acronyms in the abstract.
Image resolution of maps should be improved.
Figure 4: since the maps serve for the identification of land use changes, it might be better to color in white the areas that do not change (i.e. 11, 22, 33, 44, 55, 66). In this way it would be easier to identify those that change.
Figure 6 definitely needs higher image resolution.
Conclusions might illustrate the contribute of this study to the existing literature and might make recommendation for further research.
Author Response
Please see the attachment.

This manuscript is a resubmission of an earlier submission. The following is a list of the peer review reports and author responses from that submission.